# Global Landscape of Organic Carbon and Total Nitrogen in the Soils of Oasis Ecosystems in Southern Tunisia

**Nadhem Brahim** [1,*] , **Nissaf Karbout** [2], **Latifa Dhaouadi** [3] and **Abdelhakim Bouajila** [4]

1 Department of Geology, Faculty of Sciences of Tunis, University of Tunis El Manar, Tunis 2092, Tunisia
2 Institute of Arid Regions (IRA), Medenine 4119, Tunisia; nissaf.karbout@yahoo.fr
3 Regional Center for Research in Oasis Agriculture (CRRAO), Degache 2260, Tunisia; latifa_hydro@yahoo.fr
4 Faculty of Sciences of Gabes, University of Gabes, Gabes 6029, Tunisia; bouajilaabdelhakim@gmail.com
* Correspondence: nadhem.brahim@fst.utm.tn; Tel.: +216-23-672-384; Fax: +216-71-885-408

**Abstract:** The oasis soils of Tunisia face several climatic and soil constraints. Trying to have cultures that are profitable and beneficial in terms of soil C and N sequestration in such environments is already a challenge. To conduct this, we tested under identical conditions four types of occupation in sub-plots adjacent to the crops; barley alone, alfalfa alone, intercropping barley/alfalfa, and a control fallow in a saline gypseous desert soil poor in organic matter. Field experimentation was carried out in the oasis of Degache, which is very representative of other Tunisian oases. The stocks of C and N of the plot were calculated from the start in September 2019 before the installation of the different crops. After 21 months, the control plot shows a decrease of −41% in its stock of C and −25% in its stock N. However, the best result is that of the barley/alfalfa intercropping with an increase of +126.46% in the C stock and +178.67% in the N stock. After almost two years of experience, the beneficial effect of the intercropping system in the oasis is clear. These results are very motivating and seem to be a solution to the rapid decline in soil organic stocks.

**Keywords:** soil C and N stocks; cropping system; barley/alfalfa; saline soil; oasis; North Africa

## 1. Introduction

The fertility and sustainability of many soil systems on our planet are under threat from a diverse range of stresses put upon them, from actors such as global environmental change and pollution, loss of organic matter, erosion, salinization, reduction of available water quantity, and quality. Only few places in the world experience more of these simultaneous stresses upon their natural soil ecosystems environment than the semi-arid and arid regions of the world. These regions cover 41% of the total global land area and host 37% of the world population. These soils in the semi-arid and arid regions are characterized by low organic matter (OM) contents, yet, because of their large extent they still represent a significant global OM store.

Soil organic matter is a key factor controlling the welfare of humanity by its ability to provide agricultural resources (soil fertility and thus food) and a healthier environment at a local and global scale (e.g., attenuation of nitrate pollution and greenhouse gases emissions). However, 'sustainable' agriculture requires a clear understanding of the multiple drivers and treats to soil organic matter dynamics and associated elemental cycles in semi-arid and arid soils.

Agriculture is based on the artificial input of mineral (e.g., NPK) or organic (manure) based nutrients mainly to offset occurring nutrient loss due to crop removal to maintain agricultural productivity. However, due to the growing cost or scarcity of such resources, more emphasis will be on the 'natural' soil productivity to maintain agronomic production. Even under normal conditions after 50 to 60 years, the supply of soil available P or K may be exhausted [1]. However, mismanagement of these semi-arid lands may already have led to further degradation (erosion, salinization) of the land, which may, however, be reverted

at adapted tree re-establishment [2]. Nevertheless, erosion scours the surface soil and depletes the supply of soil nutrients, which are potentially available to crops. In addition to the challenges associated with the depletion of soil nutrients and erosion, questions arise facing biodiversity and water management. The situation is particularly dramatic in the oasis, which are the remaining fertile baskets of many arid and hyper-arid places, e.g., the Sahara Desert. Moreover, in coastal areas, the application of saline water to grow crops, such as date palms, can further enhance the salinization of soils and decrease natural productivity. Therefore, an improved understanding of the critical drivers and threats to soil organic dynamics and nutrients cycling, and thus productivity can in semi-arid and arid parts of the world have immediate direct and significant beneficial environmental and economic effects.

The oasis and its mythical tree, the date palm, are the product of a remarkable combination of man's agronomic genius and the abilities of nature's resources in a desert environment. The soil in the oases is the result of intelligent human activity. In fact, in these Saharan environments, around natural sources of freshwater, naturally soils are Lithosols and Regosols type [3], their organic matter contents, in some cases, do not exceed 0.2%. The farmers were able to "build" new "anthropogenic soils" around these water points by adding organic matter, mainly manure from their own herds of sheep, goats, and camels.

Around 900,000 Tunisians earn their living from Tunisian oases, i.e., 10% of the population [4]. In fact, the production of dates contributes 13 to 16% to tree production, 5 to 7% to plant production, and 16% to agricultural exports. As a result, Tunisia is the 4th world exporter of dates and the 1st in terms of value. The production of dates is estimated at 288 thousand tons in 2018/2019.

The oases of North Africa and precisely the Tunisian oases can be classified into three types: (i) mountainous such as those of the Gafsa mountains, (ii) coasts such as those of the coast of Gabes, and (iii) continental such as those of Sahara of Kebili or Tozeur and which are the most extensive. Today, almost all these oases are irrigated by pumped water from deep boreholes since all the springs have mostly dried up.

Currently, the life of the oases is facing the challenge of global warming. Studies predict a decrease of 17% in precipitation and an average increase in temperature of 2.7 °C by 2050 [5]. This represents a high for date palm crops due to the lack of irrigation water, the degradation of the quality of their soils through secondary salinization, and the decrease of organic matter content in soils.

In these oasis ecosystems, the continuous supply of exogenous OM for the soils is rare but essential, thus it is very useful to look for new alternatives that could help farmers maintain the good quality of their soils. Good soil quality is known by improving ecosystem services and biological productivity. In Tozeur, the agricultural development group (GDA) estimates that 1/3 of the total surface area of the abandoned oasis is threatened by desertification processes [6]. In view of the current economic situation in Tunisia, the restoration of soils with low OM content has become a strategic necessity for food security. High stocks of C and N in the soil correlate positively with soil productivity. The stock of organic matter in the soil is a key biological factor and is one of the best indicators of soil fertility [7].

Other than the additions of manure or sometimes compost that farmers added daily to these soils [8,9], the mode of tillage and mainly the vegetation sown in the herbaceous state is fundamental in its impact on the organic stock.

The diversity of plant species has a beneficial effect through both functional complementarity and facilitation between plants, which improves productivity as well as increasing organic carbon (C) and total nitrogen (N) stocks in soil, microbial biomass, and crop residues [10,11]. The effects of associated crops on the productivity of the agroecosystem and on the storage of C and N have already been verified in several studies [12]. The input of C to the soil through root residues, both for legumes and cereals, was found to be higher in intercropping systems than in monoculture systems [13]. However, such studies in oasis systems are rare or even non-existent. Farmers in the first place sought to produce

dates, and in traditional oases, some fruits with trees adapted to salt and heat stress, such as pomegranate, olive, and fig trees. Only a few have tried to produce vegetables or fodder plants for their herds. Replacing bare soil or fallow systems with profitable agricultural practices is currently a major concern of oasis farmers. With these problems in mind, a new agricultural system has been set up for the farmers of the oasis and outside their customs. It is to sow alfalfa and barley at the same time, in a nontraditional way. This could improve their forage yields for their livestock and primarily the quality of their soils. The objective of this paper is to study the effects of the different cultures encountered in the Tunisian oases, namely, fallow, varied rotations, and a new intercropping of alfalfa with barley on (i) sequestration of C and N alfalfa nodules at the layer 0–20 cm, and (ii) soil C and N stocks throughout the profile in general.

## 2. Materials and Methods

### 2.1. Study Site

The experiment was carried out in a plot located in the center of the oasis of Degache (Figure 1) (33°59′23.73″ N, 8°14′49.79″ E) with an ambitious young farmer wanting to improve his soil and his yields. The experiment took place between 2019 and 2020. For the climate, it is desert characterized by an average annual rainfall of 98.3 mm, an average annual temperature of 21.8 °C, a minimum temperature recorded in January of 6.1 °C, a maximum temperature recorded in July of 39.5 °C, a Piche evaporation 2743.3 mm/year, and an annual number of hours of insolation of 3043.4.

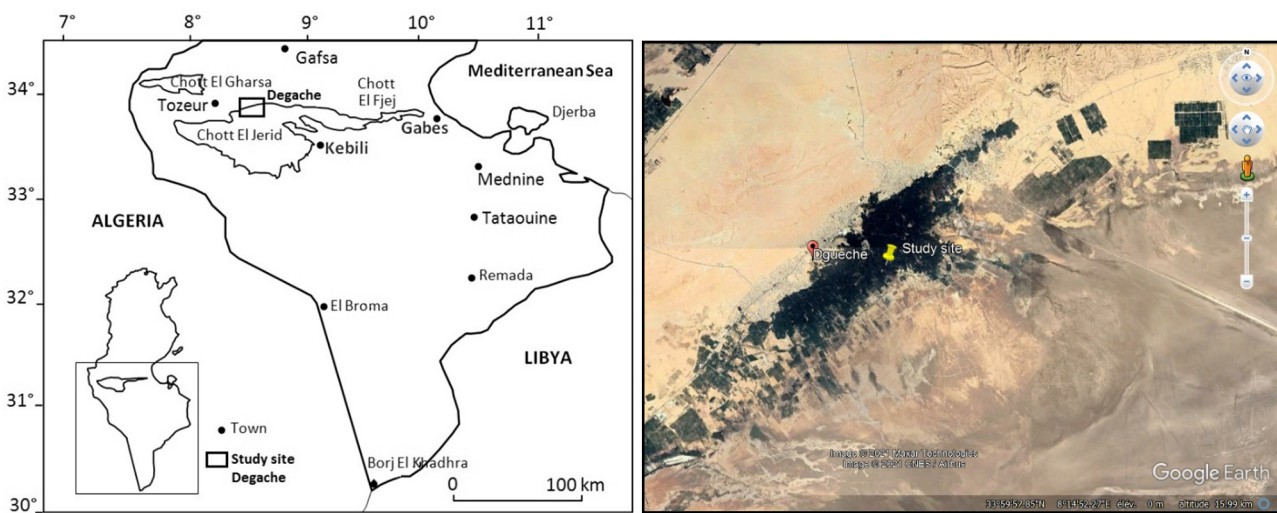

**Figure 1.** Location of the study site.

Barley (*Hordeum vulgare*) and alfalfa (*Medicago sativa*) are the 2 main fodder crops in this region. They are usually planted in the monoculture system between the trees taking advantage of the irrigation waters of the date palms. The irrigation is conventionally performed by submersion, which is from around 3000 mm/year. However, there is also the production of some seasonal vegetables such as onion, eggplant, parsley, tomato, etc. During the 2 years of study, precipitation was 15.1 mm and 52.2 mm, in 2019 and 2020, respectively.

### 2.2. Experimental Design and Soil Sampling

The farmers subdivide their plots into small sub-plots of 3 m wide by 10 m long (30 m²) in order to irrigate them with submersion. For palm dare density, each sub-plot contains 2 large date palm trees 8 to 10 m apart from each other (sometimes in between small date palm trees when they intend to rejuvenate their palm groves, after a few years remove the old trees and leave the young ones) (Figure 2). The sub-plots will be used for herbaceous plantations to take advantage of irrigation water. We chose the sub-plots as an experimental design, we performed 3 repetitions. Each sub-plot was cultivated with; (i)

barley as a sole crop (BA), (ii) alfalfa as sole crop (AL), (iii) barley-alfalfa intercrop (BA/AL), and (iv) fallow (FA: uncultivated plot) (4 cropping systems × 3 replicates).

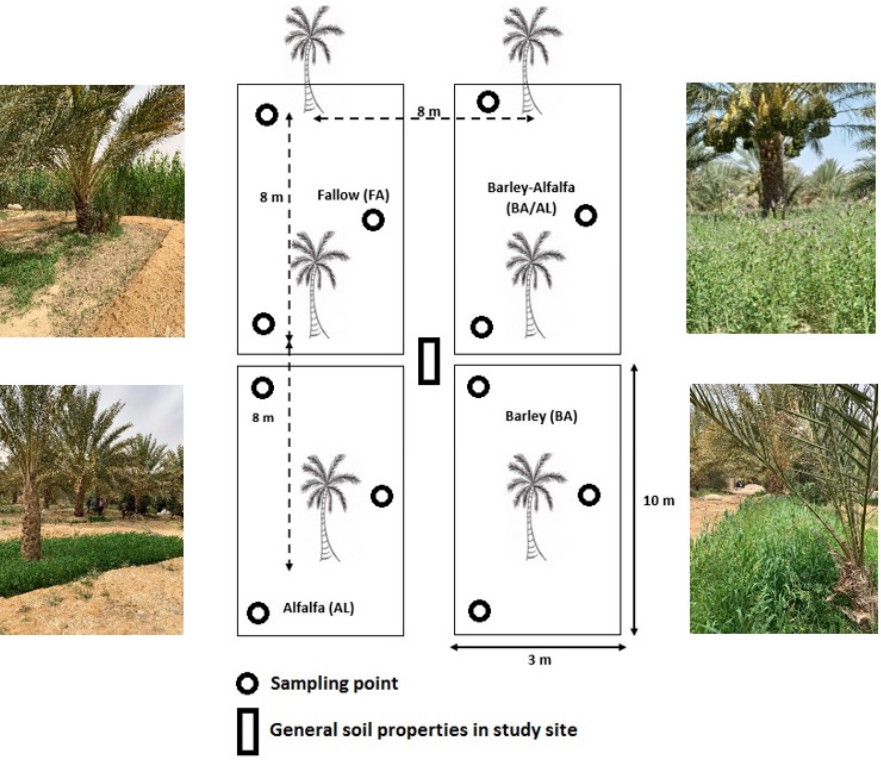

**Figure 2.** Sampling scheme at the four sub-plots in the study site.

Barley and alfalfa are planted, amended, and irrigated similarly to the farmer's method. The soil of the fallow plot was plowed and left unplanted according to local agricultural practices. The fallow soils were considered as a control.

The soil of the 4 sub-plots is prepared in the same way in August 2019 with an addition of 5 kg m$^{-2}$ of sheep and goat manure, and everything was well mixed with a deep plowing of 30 cm. After 5 months, in February 2020 an application of 100 g m$^{-2}$ of phosphate to the surface of the soil in the 4 working modes. In the second year, no amendment was added.

We used the daily additions applied by farmers in the Degache oasis at the start of each farming season. Indeed, in the customs of the oasis, for centuries, local farmers applied the addition of manure, and a few years later to the discovery of phosphate in southern Tunisia, more than a century ago (1885), farmers also applied the addition of phosphate. As for the doses per square meter, it is around 5 to 8 kg of manure and between 100 and 150 g of phosphate. Generally, this method of fertilization is applied every 3 years, with the exception of young palm groves (<4 years), where the amendment is carried out every 2 years, and in certain cases where farmers are well-off, the amendment is carried out every year.

Seeding of alfalfa and barley was carried out on 30 September 2019. The cultivation of barley ears alone was carried out on 25 April 2020 (human consumption). After 5 days, on 30 April 2020 was carried out the first cut of alfalfa and what remains of barley stubble (animal consumption). Then, every 25 days, there was an alfalfa cutout until mid-October (a total of 7 cuts as follows: 20 May 2020; 15 June 2020; 5 July 2020; 30 July 2020; 25 August 2020; 20 September 2020; and 15 October 2020). A 4-month rest for alfalfa (from 15 October 2020 until 15 February 2021). From 15 February, cutting resumed every 25 days until next October 2021. Alfalfa can remain in production for 4 years.

In 2020, the 2 alfalfa-only plots and the alfalfa-barley intercropping were already planted since 2019, the barley plot was planted again on 30 September 2020 (plowing the

soil without any additions, manure, and phosphate), and the fallow plot always fallow without any additions.

Soil samples were taken on 2 dates at the start of the agricultural year during plowing and at the end of cultivation at 25–30 May. We carried out 3 measurements over 2 years: September 2019, June 2020, and June 2021.

For soil samples, the pH was measured in the soil in purified water with a soil: water ratio of 1:2.5. The concentration of calcium carbonate ($CaCO_3$) was determined in the laboratory using the Bernard Calcimeter method. The total nitrogen concentration was determined by the Kjeldahl method, and the total organic carbon concentrations in the soil and nodules were determined by the Walkley–Black method. Analysis of major elements in the soil solution was performed by Inductively Coupled Plasma Optical Emission Spectrometry (ICP-OES) (Table 1).

### 2.3. Chemical Analysis, Soil C and N Stocks

The bulk density (BD) was determined by the cylindrical core method (BD = dry soil weight/cylinder volume). Soil organic carbon (SOC) content was measured by the oxidation method of Walkey and Black by the mixture of $K_2Cr_2O_7$–$H_2SO_4$ [14]. Total nitrogen (N) was measured by the Kjeldahl method [15]. The particle size distribution (i.e., coarse sand, fine sand, coarse silt, fine silt, and clay) reflected that of the whole soil [16]. The Gypsum percentage was determined using the ammonium carbonate treatment and precipitation by barium chloride [17]). Carbonates were analyzed according to the Scheibler method [18]. The electrical conductivity (EC) was analyzed in a saturated soil paste. The soil pH was measured in the soil in purified water with a soil: water ratio of 1:2.5. EC and pH were measured using digital pH and conductivity meters. Chemical analysis of cation elements ($Ca^{2+}$, $Mg^{2+}$, $Na^+$, and $K^+$) was performed by spectrometric analysis using an atomic absorption spectrometer. Anions ($CO_3^{2-}$, $HCO_3^{2-}$, $Cl^-$ and $SO_4^{2-}$) concentrations were estimated using standard analytical methods [19]. The SOC and soil N stocks were calculated for each individual layer using the following equations [20,21]:

$$\text{SOC (or N) stock} = \text{C (or N) content} \times \text{BD} \times \text{D} \tag{1}$$

with organic carbon (C) or total nitrogen (N) content in %, the bulk density (BD) in g cm$^{-3}$ and soil depth (D) in cm. SOC (or N) stocks are expressed in t C ha$^{-1}$ or t N ha$^{-1}$ then SOC (or N) stock converted into kg C m$^{-2}$ or kg N m$^{-2}$.

### 2.4. Soil Classification

Based on the field observation, morphological, and analytical soil properties, the studied soil profile was classified using soil WRB as Gypsisols [22]. Indeed, the studied soil contained an accumulation of gypsum in depth. These soils, by definition, are located in the driest parts of the desert climatic zone with inevitably a gypsum encrustation ($CaSO_4 2H_2O$).

### 2.5. Statistical Analysis

Data were analyzed using the statistical analysis software (IBM SPSS Statistics Version 20, Chicago, IL, USA). Analysis of variance was conducted to test the significance of treatments, soil depth, particle size, and their interaction regarding soil organic carbon and total nitrogen stocks.

**Table 1.** General soil properties in study site, September 2019 (n = 3).

| D (cm) | Particle Size (%) Sand | Particle Size (%) Silt | Particle Size (%) Clay | pH | C (%) | N (%) | BD (g/cm) | C Stock kg C m$^{-2}$ | N Stock kg N m$^{-2}$ | EC (mS/cm) | CaCO$_3$ (%) | Gypsum (%) |
|---|---|---|---|---|---|---|---|---|---|---|---|---|
| 0–15 | 86.20 ± 1.10 | 10.00 ± 1.15 | 3.80 ± 1.01 | 7.80 ± 0.29 | 1.33 ± 0.788 | 0.084 ± 0.02 | 1.57 ± 0.003 | 3.14 ± 0.182 | 0.20 ± 0.005 | 4.70 ± 0.20 | 7.00 ± 0.72 | 10.00 ± 1.36 |
| 15–30 | 84.70 ± 2.19 | 13.10 ± 2.80 | 2.20 ± 0.96 | 7.70 ± 0.25 | 1.12 ± 0.020 | 0.078 ± 0.03 | 1.55 ± 0.003 | 2.59 ± 0.041 | 0.18 ± 0.007 | 4.80 ± 0.29 | 22.00 ± 3.52 | 13.00 ± 1.97 |
| 30–45 | 80.80 ± 2.31 | 8.90 ± 3.20 | 10.30 ± 1.32 | 7.70 ± 0.26 | 0.81 ± 0.038 | 0.077 ± 0.02 | 1.62 ± 0.006 | 1.96 ± 0.087 | 0.19 ± 0.005 | 5.70 ± 0.10 | 26.00 ± 1.89 | 24.00 ± 1.61 |
| 45–60 | 65.60 ± 1.85 | 17.20 ± 2.26 | 17.20 ± 2.06 | 7.80 ± 0.15 | 0.33 ± 0.008 | 0.070 ± 0.05 | 1.72 ± 0.003 | 0.85 ± 0.024 | 0.18 ± 0.012 | 6.80 ± 0.42 | 18.00 ± 2.93 | 47.00 ± 1.15 |

±: St. E.

## 3. Results

### 3.1. Pre-Experiment Soil Characteristics, September 2019

The soil of the four sub-plots was prepared in September 2019 (addition of manure and plowing). The pedological nature of the four sub-plots remained the same: a gypsum soil belonging to the class of Gypsisols. Then, a soil pit was made in the center of the four sub-plots to determine the general characteristics of the soil of the study site (Figure 2). The results of the analysis carried out on the soil, and the soil solution are detailed in Tables 1 and 2.

According to the granulometric analysis, the soil texture was sandy loam down to a depth of 30 cm. Beyond that, it was sandy clay. The texture of oasis soils was characterized by the abundance of the sandy fraction (>60%). This is a normal result since oases originate in the desert. In 2019, the BD was about 1.57 g/cm$^3$ at the surface layer (0–15 cm) and 1.72 g/cm$^3$ in depth layer (45–60 cm). The soil organic carbon content was around 1.33% on the surface and decreased with depth layer to reach 0.33% at 60 cm depth. However, soil total nitrogen content was almost the same at the surface, and at depth, the values varied between 0.08 and 0.07% (Table 1).

The soil was characterized by its alkalinity, with a pH between 7.7 and 7.8. It was also carbonated, the HCl test was positive throughout the profile. Moreover, in our soil, the CaCO$_3$ content was between 7 and 26%. The gypsum concentration increased with depth, ranging from 10% on the surface (0–15 cm) up to 47% in the 45–60 cm layer. This increase in the gypsum content characterizes oasis soils until the formation of a gypsum slab to a depth of 1.5 m [8]. The electrical conductivity (EC) shows saline soil, the values of EC were between 4.7 and 6.8 mS/cm. The analysis of soluble salts in the soil solution shows the abundance of the elements Na$^+$ and Cl$^-$ especially in depth. Moreover, the sodium adsorption ratio (SAR) increased with the depth, confirming this increasing salinity with the depth. The analysis shows that the SAR values go from 4.1 to 4.3 in the two surface layers (0–15 and 15–30 cm) to 10.7 and 13.7, respectively, in the layers 30–45 and 45–60 cm. This shows that the soil is saline and that the salinity increases with the depth (Table 2).

### 3.2. Soil pH, BD, EC, C, and N in 2020 and 2021

The pH showed a variation according to the vegetation during the two years of the experiment 2020 and 2021 compared to the reference year 2019. Indeed, as already indicated in Figure 3, the variation is quite clear under the intercropping of barley and alfalfa (BA/AL), where pH values in comparison with the reference year 2019 have decreased. This decrease was very clear in the year 2021 in the 30–45 cm layer, where the pH drops from 7.73 in 2019 to 7.23 in 2021 (Tables 3 and S1).

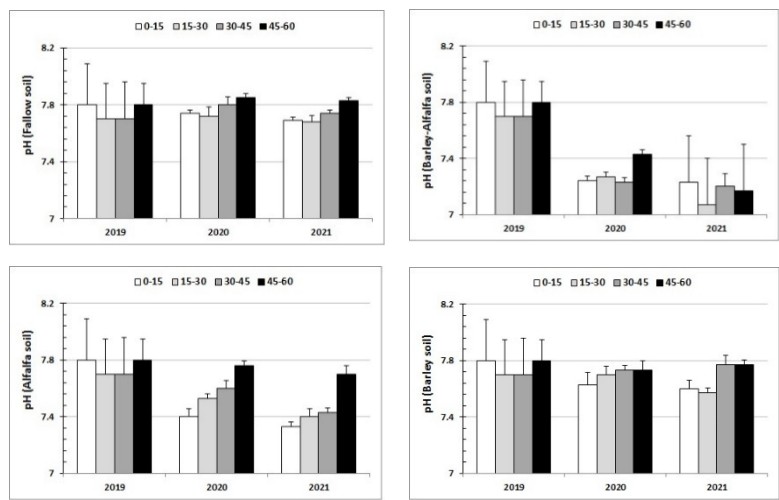

**Figure 3.** Variation of soil pH according to the vegetation in 2020 and 2021.

**Table 2.** Major elements of soil solution in the study site, September 2019 (n = 3).

| D (cm) | Soluble Salts of the Soil Solution Extract (1/5) mEq/L | | | | | | | | SAR |
|---|---|---|---|---|---|---|---|---|---|
| | $Ca^{2+}$ | $Mg^{2+}$ | $Na^+$ | $K^+$ | $CO_3^{2-}$ | $HCO_3^-$ | $Cl^-$ | $SO_4^{2-}$ | |
| 0–15 | 30.80 ± 1.59 | 4.20 ± 0.45 | 17.20 ± 0.81 | 0.30 ± 0.06 | 0.02 ± 0.01 | 0.20 ± 0.06 | 17.10 ± 2.72 | 43.20 ± 3.96 | 4.11 ± 0.16 |
| 15–30 | 27.10 ± 2.53 | 8.30 ± 0.21 | 18.50 ± 2.32 | 1.10 ± 0.17 | 0.01 ± 0.00 | 0.40 ± 0.12 | 20.10 ± 1.35 | 37.20 ± 3.01 | 4.39 ± 0.41 |
| 30–45 | 42.80 ± 1.14 | 55.20 ± 2.75 | 75.20 ± 5.61 | 18.80 ± 2.41 | 0.02 ± 0.01 | 0.80 ± 0.21 | 99.10 ± 7.85 | 192.20 ± 2.52 | 10.74 ± 0.78 |
| 45–60 | 32.60 ± 2.09 | 20.00 ± 1.67 | 70.20 ± 0.38 | 4.20 ± 0.55 | 0.01 ± 0.00 | 0.40 ± 0.06 | 122.00 ± 1.53 | 44.90 ± 7.70 | 13.69 ± 0.48 |

**Table 3.** Soil properties at the four sub-plots between 2020 and 2021 (n = 3).

| D (cm) | pH | | EC (mS/cm) | | C (%) | | N (%) | | C/N | | BD (g/cm³) | |
|---|---|---|---|---|---|---|---|---|---|---|---|---|
| | 2020 | 2021 | 2020 | 2021 | 2020 | 2021 | 2020 | 2021 | 2020 | 2021 | 2020 | 2021 |
| **Barley (BA)** | | | | | | | | | | | | |
| 0–15 | 7.63 ± 0.088 | 7.60 ± 0.058 | 3.80 ± 0.794 | 4.03 ± 0.867 | 1.44 ± 0.005 | 1.38 ± 0.009 | 0.09 ± 0.006 | 0.07 ± 0.006 | 16.133 ± 1.099 | 19.93 ± 1.534 | 1.477 ± 0.015 | 1.48 ± 0.003 |
| 15–30 | 7.70 ± 0.057 | 7.57 ± 0.033 | 3.67 ± 0.994 | 3.43 ± 0.994 | 1.126 ± 0.014 | 1.08 ± 0.003 | 0.08 ± 0.006 | 0.07 ± 0.003 | 14.2 ± 0.839 | 16.33 ± 0.933 | 1.507 ± 0.003 | 1.54 ± 0.003 |
| 30–45 | 7.73 ± 0.033 | 7.77 ± 0.067 | 3.30 ± 0.458 | 3.60 ± 01.026 | 0.88 ± 0.011 | 0.86 ± 0.006 | 0.087 ± 0.003 | 0.08 ± 0.003 | 10.2 ± 0.416 | 10.37 ± 0.393 | 1.503 ± 0.006 | 1.55 ± 0.003 |
| 45–60 | 7.73 ± 0.066 | 7.77 ± 0.033 | 5.07 ± 0.441 | 3.33 ± 01.135 | 0.316 ± 0.008 | 0.45 ± 0.006 | 0.06 ± 0.006 | 0.08 ± 0.003 | 5.4 ± 0.666 | 5.90 ± 0.208 | 1.703 ± 0.003 | 1.71 ± 0.006 |
| **Alfalfa (AL)** | | | | | | | | | | | | |
| 0–15 | 7.40 ± 0.058 | 7.33 ± 0.033 | 4.23 ± 0.033 | 3.83 ± 0.033 | 1.45 ± 0.006 | 1.56 ± 0.003 | 0.12 ± 0.006 | 0.14 ± 0.009 | 12.13 ± 0.581 | 10.93 ± 0.669 | 1.46 ± 0.003 | 1.45 ± 0.006 |
| 15–30 | 7.53 ± 0.033 | 7.40 ± 0.058 | 4.30 ± 0.000 | 3.73 ± 0.203 | 1.06 ± 0.029 | 1.39 ± 0.033 | 0.10 ± 0.007 | 0.20 ± 0.012 | 10.33 ± 0.987 | 7.00 ± 0.306 | 1.51 ± 0.012 | 1.53 ± 0.006 |
| 30–45 | 7.60 ± 0.057 | 7.43 ± 0.033 | 4.90 ± 0.058 | 4.77 ± 0.088 | 1.06 ± 0.031 | 0.81 ± 0.047 | 0.13 ± 0.012 | 0.18 ± 0.009 | 8.27 ± 0.623 | 4.47 ± 0.467 | 1.53 ± 0.003 | 1.66 ± 0.003 |
| 45–60 | 7.76 ± 0.033 | 7.70 ± 0.058 | 5.57 ± 0.033 | 5.13 ± 0.033 | 0.45 ± 0.009 | 0.43 ± 0.018 | 0.09 ± 0.006 | 0.10 ± 0.010 | 5.07 ± 0.273 | 4.40 ± 0.404 | 1.71 ± 0.012 | 1.71 ± 0.006 |
| **Barley-Alfalfa (BA/AL)** | | | | | | | | | | | | |
| 0–15 | 7.24 ± 0.032 | 7.23 ± 0.333 | 3.83 ± 0.033 | 2.43 ± 0.033 | 1.87 ± 0.006 | 3.96 ± 0.078 | 0.14 ± 0.003 | 0.33 ± 0.006 | 13.70 ± 0.300 | 12.00 ± 0.100 | 1.44 ± 0.003 | 1.42 ± 0.012 |
| 15–30 | 7.27 ± 0.033 | 7.07 ± 0.333 | 3.80 ± 0.000 | 2.40 ± 0.058 | 1.64 ± 0.019 | 2.51 ± 0.330 | 0.20 ± 0.009 | 0.23 ± 0.006 | 8.37 ± 0.418 | 11.00 ± 1.701 | 1.44 ± 0.023 | 1.43 ± 0.007 |
| 30–45 | 7.23 ± 0.033 | 7.20 ± 0.088 | 4.30 ± 0.078 | 3.33 ± 0.233 | 1.37 ± 0.006 | 1.44 ± 0.003 | 0.18 ± 0.003 | 0.20 ± 0.015 | 7.50 ± 0.153 | 7.40 ± 0.586 | 1.54 ± 0.020 | 1.50 ± 0.031 |
| 45–60 | 7.43 ± 0.033 | 7.17 ± 0.333 | 4.73 ± 0.067 | 3.80 ± 0.058 | 0.78 ± 0.006 | 0.88 ± 0.012 | 0.17 ± 0.003 | 0.18 ± 0.015 | 4.50 ± 0.058 | 5.07 ± 0.498 | 1.67 ± 0.047 | 1.71 ± 0.101 |
| **Fallow (FA)** | | | | | | | | | | | | |
| 0–15 | 7.74 ± 0.023 | 7.69 ± 0.021 | 4.30 ± 0.057 | 4.23 ± 0.913 | 0.82 ± 0.035 | 0.72 ± 0.026 | 0.08 ± 0.000 | 0.05 ± 0.001 | 10.43 ± 0.470 | 14.27 ± 0.353 | 1.57 ± 0.019 | 1.57 ± 0.015 |
| 15–30 | 7.72 ± 0.062 | 7.68 ± 0.042 | 4.26 ± 0.417 | 4.07 ± 0.636 | 0.61 ± 0.024 | 0.64 ± 0.015 | 0.06 ± 0.000 | 0.06 ± 0.003 | 10.10 ± 0.265 | 11.30 ± 0.800 | 1.54 ± 0.003 | 1.58 ± 0.038 |
| 30–45 | 7.80 ± 0.053 | 7.74 ± 0.022 | 4.76 ± 0.133 | 4.37 ± 0.524 | 0.52 ± 0.012 | 0.44 ± 0.015 | 0.07 ± 0.003 | 0.06 ± 0.004 | 7.87 ± 0.593 | 7.30 ± 0.306 | 1.61 ± 0.006 | 1.65 ± 0.040 |
| 45–60 | 7.85 ± 0.026 | 7.83 ± 0.018 | 6.76 ± 0.491 | 6.07 ± 0.371 | 0.29 ± 0.023 | 0.29 ± 0.021 | 0.06 ± 0.006 | 0.06 ± 0.001 | 4.87 ± 0.684 | 4.87 ± 0.333 | 1.72 ± 0.009 | 1.72 ± 0.015 |

The EC varies according to Table 3 from 3.30 to 6.76 mS/cm for the year 2020 and from 2.4 to 6.07 mS/cm for the year 2021. The variation was mainly explained by the effect of leaching salts from the soil by irrigation water or by the capillary rise of drainage water from the oasis. However, a clear decrease in EC was clear in the vegetated soils compared to the FA subplot. The lowest EC values were recorded in the barley/alfalfa intercropping plot (BA/AL). At the 45–60 cm layer, the deepest in our study, we recorded the highest EC values encountered on all plant occupations. This is explained by the increase in salinity as a function of depth, thus the gypsum content increases as a function of depth. The only exception was found in the BA subplot, where the 45–60 cm layer had the lowest EC value in comparison with the other surface layers (Figure 4).

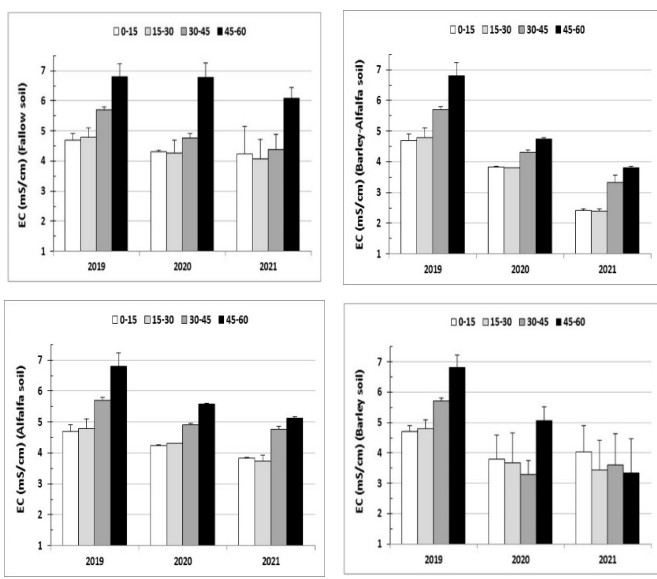

**Figure 4.** Variation of soil EC according to the vegetation in 2020 and 2021.

In 2019, the reference profile presented values of BD ranging from 1.57 to 1.72 g/cm$^3$. In 2020, the lowest BD values were near the surface horizons of the barley-alfalfa association subplot (BA/AL) with a value of 1.44 g/cm$^3$. The decrease was also at the level of the layer deep 45–60 cm with a value of 1.67 g/cm$^3$. In 2021, once again, the intercropping barley-alfalfa (BA/AL) sub-plot showed the lowest values of BD, ranging from 1.42 g/cm$^3$ in 0–15 cm to 1.43 g/cm$^3$ in the 15–30 cm layer. This is indicative of an improvement in porosity and an increase in the organic matter content of the soil (Table 3).

Figure 5 shows the general appearance of the BD as a function of depth, the first observation is that the BD values decrease as a function of depth and the value of the BD in the layer 45–60 cm is always high around 1.7 g/cm$^3$. The associated BA/AL culture remains the best from the point of view of improvement of the BD, followed by the AL culture alone, then finally the BA culture alone.

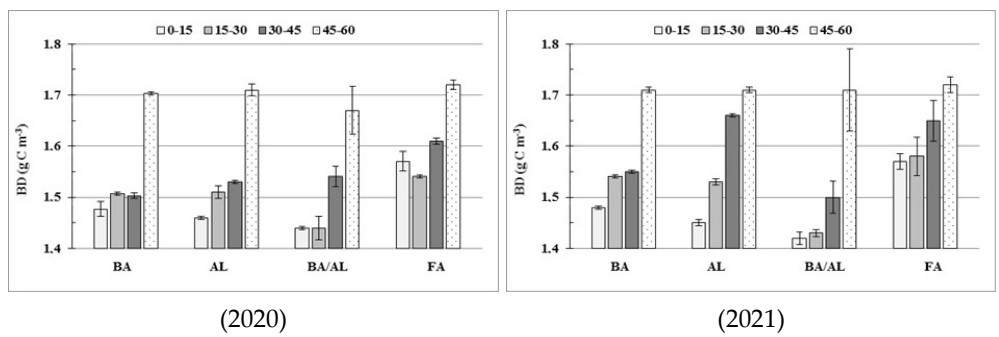

(2020)  (2021)

**Figure 5.** Evolution of the BD as a function of the depth in the soils of the four types of vegetation.

Regarding the content of C and N, overall, an improvement was recorded in the cultivation plots of BA, AL, and BA/AL. For the organic carbon content, the best increase was observed in the intercropping BA/AL sub-plot. In fact, in the surface layer (0–15 cm), the content was of the order of 1.87% in 2020 and increased further in 2021 to reach a value of 3.96% in 2021. This improvement was really throughout the profile down to the depth of 45–60 cm.

What was observed with the organic carbon contents was the same as with the total nitrogen content. The best increases in N content were with soils of the intercropping of barley and alfalfa (BA/AL) with levels of 0.14% in 2020 and 0.33% in 2021, respectively. Alfalfa cultivation alone showed an increase but did not exceed 0.14%, half of what exists in the soil of the BA/AL intercropping for the same 0–15 cm layer (Table 3). However, the lowest nitrogen content was with barley alone, where the content did not exceed 0.07%, a value very close to fallow for the same 0–15 and 15–30 cm layer.

Overall, the C/N values in 2020 were between 10.43 and 16.13 at the surface and between 4.5 and 5.4 at depth, respectively. In 2021, the values were between 10.93 and 19.93 at the surface and between 4.4 and 5.07 at depth, respectively. These results clearly show that the mineralization of organic matter is much faster on the surface in comparison with the deep layer, where the organic matter appears to be poorly mineralized (Table 3).

*3.3. Carbon Stocks*

During the two years, SOC and soil N stocks were calculated in all four sub-plots; the fallow (FA), the sole crop BA, the sole crop AL, and the intercropping BA/AL.

For the C stock in June 2020, the analysis showed that the two unique crops of barley and alfalfa had very close stocks, with 8.57 and 9.15 kg C m$^{-2}$, respectively. The intercropping barley-alfalfa sub-plot showed a higher stock in the first three layers 0–15, 15–30, and 30–45 cm compared to their analogs of all other sole crops and fallow. In the second year of 2021, the storage of C in the nodule zone (0–15 and 15–30 cm) was significantly higher in intercropping culture than in sole culture. Figure 6 clearly shows this difference in the sequestration of C.

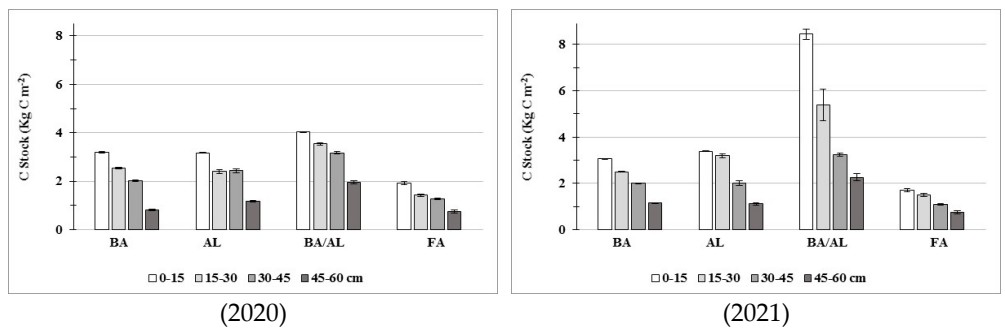

|   (2020)   |   (2021)   |

**Figure 6.** Evolution in organic carbon stocks in 2020 and 2021 under the different vegetations.

If we look at the total stock at 0–60 cm depth we have the following ranking for the year 2020; "FA", the "sole crop BA", "the sole crop AL", and "the intercropping BA/AL" with stocks respectively, 5.35; 8.57; 9.15 and 12.7 kg C m$^{-2}$. A year later, in 2021, we still have the same order of classification between the sub-plots, respectively, with 5.04; 8.71, 9.71, and 19.34 kg C m$^{-2}$ (Table 4). With reference to the stock of 8.54 kg C m$^{-2}$ for the year 2019, an overall decrease in the stock was recorded in FA of the order of −3.5%, small sequestration of C in the single-crop plots +1.99% under BA and 13.70% under AL, but a significant jump in the stock under BA/AL of 126.46% (Table 5).

In the first year, the stock of C seemed constant under sole crops BA, with a slight improvement in sole crop AL crop and with a 50% improvement in intercropping BA/AL. In the second year, sole crop barley and alfalfa only contributed 13.7%. However, with intercropping, the system appears to be very powerful in sequestering carbon.

**Table 4.** Soil organic carbon and soil total nitrogen stocks at the four sub-plots between 2020 and 2021.

| | Barley (BA) | | Alfalfa (AL) | | Barley-Alfalfa (BA/AL) | | Fallow (F) | |
|---|---|---|---|---|---|---|---|---|
| | kg C m$^{-2}$ | kg N m$^{-2}$ | kg C m$^{-2}$ | kg N m$^{-2}$ | kg C m$^{-2}$ | kg N m$^{-2}$ | kg C m$^{-2}$ | kg N m$^{-2}$ |
| June 2020 | | | | | | | | |
| 0–15 | 3.19 ± 0.019 | 0.20 ± 0.015 | 3.17 ± 0.014 | 0.26 ± 0.013 | 4.03 ± 0.021 | 0.30 ± 0.008 | 1.93 ± 0.061 | 0.19 ± 0.004 |
| 15–30 | 2.55 ± 0.038 | 0.18 ± 0.013 | 2.39 ± 0.078 | 0.23 ± 0.014 | 3.54 ± 0.042 | 0.43 ± 0.017 | 1.42 ± 0.054 | 0.14 ± 0.001 |
| 30–45 | 2.02 ± 0.024 | 0.20 ± 0.007 | 2.43 ± 0.072 | 0.30 ± 0.027 | 3.17 ± 0.051 | 0.42 ± 0.009 | 1.26 ± 0.032 | 0.16 ± 0.010 |
| 45–60 cm | 0.81 ± 0.023 | 0.15 ± 0.015 | 1.16 ± 0.029 | 0.23 ± 0.016 | 1.96 ± 0.068 | 0.44 ± 0.019 | 0.74 ± 0.062 | 0.16 ± 0.016 |
| Total | 8.57 ± 0.104 | 0.73 ± 0.05 | 9.15 ± 0.193 | 1.02 ± 0.07 | 12.7 ± 0.182 | 1.59 ± 0.053 | 5.35 ± 0.209 | 0.65 ± 0.031 |
| June 2021 | | | | | | | | |
| 0–15 | 3.06 ± 0.022 | 0.16 ± 0.013 | 3.39 ± 0.020 | 0.31 ± 0.018 | 8.44 ± 0.229 | 0.70 ± 0.016 | 1.70 ± 0.076 | 0.12 ± 0.003 |
| 15–30 | 2.50 ± 0.011 | 0.15 ± 0.008 | 3.20 ± 0.088 | 0.46 ± 0.028 | 5.39 ± 0.682 | 0.50 ± 0.014 | 1.51 ± 0.067 | 0.13 ± 0.004 |
| 30–45 | 2.00 ± 0.017 | 0.19 ± 0.008 | 2.01 ± 0.114 | 0.46 ± 0.022 | 3.25 ± 0.062 | 0.44 ± 0.041 | 1.08 ± 0.040 | 0.15 ± 0.006 |
| 45–60 cm | 1.15 ± 0.017 | 0.20 ± 0.009 | 1.11 ± 0.044 | 0.26 ± 0.027 | 2.26 ± 0.154 | 0.45 ± 0.029 | 0.75 ± 0.058 | 0.16 ± 0.004 |
| Total | 8.71 ± 0.067 | 0.7 ± 0.04 | 9.71 ± 0.266 | 1.49 ± 0.095 | 19.34 ± 1.127 | 2.09 ± 0.1 | 5.04 ± 0.241 | 0.56 ± 0.017 |

**Table 5.** Evolution of C and N stocks (0–60 cm) in the four sub-plots after two years.

| | C Stock (kg C m$^{-2}$) | | | | | N Stock (kg N m$^{-2}$) | | | | |
|---|---|---|---|---|---|---|---|---|---|---|
| | 2019 | 2020 | 2021 | 2021–2019 | Stock Evolution % | 2019 | 2020 | 2021 | 2021–2019 | Stock Evolution % |
| BA | 8.54 | 8.57 | 8.71 | 0.17 | +1.99 | 0.75 | 0.73 | 0.70 | −0.05 | −6.67 |
| AL | 8.54 | 9.15 | 9.71 | 1.17 | +13.70 | 0.75 | 1.02 | 1.49 | 0.74 | +98.67 |
| BA/AL | 8.54 | 12.7 | 19.34 | 10.8 | +126.46 | 0.75 | 1.59 | 2.09 | 1.34 | +178.67 |
| FA | 8.54 | 5.35 | 5.04 | −3.5 | −40.98 | 0.75 | 0.65 | 0.56 | −0.19 | −5.33 |

### 3.4. Nitrogen Stocks

Total nitrogen stocks in the different sub-plots followed the same trend of organic carbon stocks. The N stocks in 2020 were in the sub-plots of barley, alfalfa, barley/alfalfa, and fallow, respectively, 0.73, 1.02, 1.59, and 0.65 kg N $m^{-2}$, and in 2021; 0.7, 1.49, 2.09, and 0.56 kg N $m^{-2}$. In terms of soil nitrogen sequestration, the sole crop sub-plots were better than the fallow sub-plot, but less than the intercropping barley and alfalfa sub-plot (Figure 6). In the second year (2021), the stock under barley (BA) alone seems to be declining and approaching that of the fallow (FA). The sole crop AL and the intercropping BA/Al stocks in the different layers were constantly increasing (Table 4).

With reference to the stock of 0.75 kg N $m^{-2}$ for the year 2019 at the start of the experiment and within the entire profile (0–60 cm), after 9 months in June 2020, the sub-plot of BA seems to have almost the same stock (0.73 kg N $m^{-2}$). The stock of the control sub-plot (fallow) decreased (0.65 kg N $m^{-2}$), only the stocks of sole crop sub-plot AL and intercropping BA/Al were increasing with, respectively, 1.02 and 1.59 kg N $m^{-2}$. After one year in June 2021, the same observation was confirmed, the BA sub-plot lost 6.67% of its initial stock, the FA sub-plot also lost 25.33% of its initial stock. However, the nitrogen stock of the AL sub-plot increased by 98.67%, and that of intercropping BA/AL sub-plot increased by 178.67% (Table 5).

## 4. Discussion

The pH in the soils of the oasis of Degache was between 7.2 and 7.7. These values were due to the high salt content in the soil layers coming from the gypsum crust and to the capillary rise of drainage water from the oasis. Our results corroborate with other results found in Tunisian mainland oases characterized by an alkaline pH ranging from 7.5 to 8.5 [8,23–26]. The soil pH of the oasis of Degache is even close to the pH of some oases in Egypt. Most pH soil profiles had pH values above 7.0 and ranged from 7.31 to 9.33 in all layers [27]. Geographically, the oases are located at the level of the deserts where the soils belong to one of the two classes of Lithosols or Regosols are known by a low organic matter of the order of 0.5% [3,8,28] However, the soils of the oases are anthropogenic soils [9,23]. Human intervention increases the OM content and makes these soils very similar to other soils under humid and rainy bioclimates [26].

In the 0–15 cm layer, the four sub-plots had an organic carbon content of 1.33% and a total nitrogen content of 0.084%. This is explained by the continual addition of organic amendments generally in the form of manure [9]. However, in the "FA" fallow control plot where the farmer added nothing for his soil for 21 months (from 30 September 2019 to 30 June 2021), there was a clear drop in the C and N content for the same surface layer (0–15 cm) of 0.72 and 0.05%, respectively. The biodegradation of organic compounds depends on the nature of the organic inputs, simple compounds, whole debris, organo-mineral complexes [29,30].

The soil texture at our study site is very sandy. The sand content in the first three layers 0–15, 15–30, and 30–45 cm is 86.2, 84.80, and 80.80%, respectively. This kind of texture cannot preserve organic matter for the following reasons; (i) enough clay is needed to save OM in their sheets [31], (ii) very high heat in oasis areas requires continuous irrigation of the palm grove at a rate of four to twice a month, especially when we know that the irrigation by submersion can reach the equivalent of 3000 mm/year [32]. Consequently, the C and N will be leached and easily transported by water. This result has been observed in sandy soils in China [33], and (iii) soil salinity and abnormal conditions favor the destruction of C and N and the non-conservation of the latter [34].

The variation in N content in the four sub-plots follows the same evolution of C. The concentration was highest in the upper layers of the soil and gradually decreased with the depth of the soil. The values of the N content in our oasis were of the order of 0.33% for the richest layers and up to 0.06% in the poorest layers. This result corroborates with what has been observed in other Tunisian continental oases in Kebili [32,34,35] and even in the maritime oases of Gabes [25].

The EC of the soil samples studied ranged from 4.7 to 6.80 mS/cm in September 2019, with the highest values recorded in June 2020 of 3.80, 5.57, 4.73, and 6.07 mS/cm, respectively, in the different sub-plots; BA, AL, BA/AL, and FA (Table 3). In June 2021, the values decreased under the BA/AL association, where we recorded the lowest value of 2.4 mS/cm. The variability of soil salinity in all soil layers was high and increased with depth. This variation is because the surface layer is more sensitive to climatic conditions and agricultural management practices than the other layers. Previously published work has concluded that EC in soil influences vegetation distribution, especially in arid and semi-arid regions [36,37]. The EC is very sensitive to soil moisture, indeed an irrigation or a rain shower can change the soil salinity, which explains why the variation of the EC at the plot level of intercropping BA/AL vegetation development is also positively correlated with EC, vegetation does not have the same tolerance to environmental saline stress [27].

The decrease in the values of the BD in the sub-plots of sole crop BA, sole crop AL, and intercropping BA/AL is explained by the decrease in the compaction of the soil when it is planted, but soils bulk densities are softened due to the fine root carpet, very dense in the case of sole crop AL and intercropping BA/AL. The values of the BD under the intercropping BA/AL go on the surface (0–15 cm) from 1.57 g/cm$^3$ and in depth (45–60 cm) from 1.72 to, respectively, 1.42 and 1.71 g/cm$^3$ between September 2019 and June 2021. In most cases, microbial and arthropod activity leads to soil aeration [38]. The soils of the agroecosystems of alfalfa in association with barley. One of the main indicators of an improvement in the organic stock in our soil is the decrease in the values of its BD, which influences the organic matter content and consequently the organic carbon content of the soil [39].

Some others explain this organic carbon and total nitrogen content in the surface layers, especially in the 0–20 cm layer, by the effect of salinity, which plays a limiting and inhibiting role in the rapid mineralization of organic matter, which remains protected from decomposing microorganisms under the effect of salt stress [40,41].

Most of the soils in this study area are characterized by high sand contents and low organic C contents. The low clay content and the lack of aggregation on the surface make the soils vulnerable and even more so the crops always seeking to satisfy their need for water to survive. Coarse is dominant in the soils of our study site, and too little clay discourages aggregation and constitutes the first constraint to the sequestration of organic carbon and total nitrogen [42]. Moreover, farmers find themselves obliged to continually add manure in order to ensure the minimum amount of organic matter in their soils is conducive to making them productive [9].

Initially, after its preparation for the 2019–2020 agricultural season in September 2019, the soil of the study site contains a stock of C in the different layers 0–15, 15–30, 30–45, and 45–60 cm, respectively, of 3.14, 2.59, 1.96 and 0.85 kg C m$^{-2}$, it is for the nitrogen stock 0.20, 0.18, 0.19, and 0.18 kg N m$^{-2}$. The total stock over 0–60 cm is of the order of 8.54 kg cm$^{-2}$ for the stock of C and 0.75 kg N m$^{-2}$ for the stock of N. These values are higher compared to the values calculated by Omar et al. (2017) [24] in the 0–30 cm layer in other soils and other Tunisian oases where the stock was of the order of 2.45 kg C m$^{-2}$ in a Solonchak at the Gataya oasis and 3.19 in a Solonchak from the oasis Ras El Ain, against 5.73 kg C m$^{-2}$ in our study. This is explained by the times of soil sampling by the authors, September is the month of land preparation and addition of manure.

By comparing the stock of carbon or nitrogen in our study site for the month of September 2019 with that of the fallow plot in June 2021, the loss of the organic stock is evident. It is of the order of −40% for the stock of C and −25.33% for the stock of N. This result shows the fragility of the stock of C and N in oasis soils where all the conditions are unfavorable; the low content of clay, high temperature, leaching by irrigation and abundance of salts. The stock calculation should be pushed to significant depths of 1.5 or 2 m, the soils are sandy, and leaching with irrigation is very active in front of such a sandy texture and lumpy structure.

Thus, the sandy texture of the soil, application of manure, and irrigation can be another reason why organic carbon and total nitrogen increased and accumulated in the subsoil after cultivation. These are also potential reasons why the soil organic carbon and the microbial biomass carbon will increase in the deep layers, as in the case of the oases of China [41,43].

The integration of the adequate annual crop seems a relevant and adequate solution to the lack of manure and its cost for small farmers who do not have animal husbandry in parallel.

After comparing the stocks from the barley crop alone (BA), the alfalfa crop alone (AL), and the barley/alfalfa intercropping crop (BA/AL), the C and N content in June 2021 increased considerably in the surface layer and up to the depth of 60 cm over a short period, with the following stock ratios, respectively: +1.99, +13.70 and +126.46% for the C stock and for the nitrogen stock −6.67, +98.67 and +178.67%. Our results are consistent with other similar research findings [42,44–46], indicating that the change in land use from annual crops to perennial grasses increases soil organic carbon levels by sequestering C in the soil profile.

Among the reasons for the increase in organic stock precisely in the alfalfa plot alone or in barley/alfalfa intercropping was the lack of tillage, which, among other things, preserves the organic stock in the surface layers, it is the elimination of tillage [47] and good root mass from alfalfa compared to other plants [48].

The C and N stock increased as a result of BA/AL intercropping. This is also a result of soil type, soil texture, the abundance of nutrients in that soil, the effect of the topography, the degree of its salinity, and other variables, which can influence the organic sequestration. In terms of the order of magnitude of the C stock following alfalfa cultivation, all the values showed an increase in the stock but not the same value. Moreover, several authors showed that if we set the same variables such as the type of crop and the initial treatment with fertilizer, C and N stocks do not change in the same order of magnitude [24,42].

It is, therefore, very useful to conduct an in-depth study on the bacterial community, the microbial carbon and nitrogen in the roots, to also study the respiration of this soil under the effect of the different cultures BA, AL, BA/AL, and FA. Certainly, this improvement in the stock of C and N has effects on the salinity, which manifests itself in the variation of the values of pH and EC also necessitates a deep study of the four sub-plots. Studies have shown a fungal and microbial dominance in heterotrophic respiration in soil outside oases, against a dominance of increased bacterial respiration in oasis soil [41]. To address these critical questions, there is a clear need to carry out long-term studies on the effects of land-use change on carbon storage in soil oasis.

Latati et al. (2017) [43] showed a similar result with the application of intercropping maize and common bean in northern Algeria. We must also follow the bacterial C stock as well as its evolution during bacterial respiration in order to understand the effect of C sequestration at the rhizosphere. Studies of the bacterial carbon stock, as well as its evolution during bacterial respiration, have been carried out in humid and semi-arid Mediterranean climates [43,49,50]. However, in this Mediterranean desert context, we did not really know. In calcareous Mediterranean soils, the microbial parameters of stocks (of C and N) and their links with environmental variables have been studied [51], we must follow the same research paths with Mediterranean gypsum and saline soils to better understand the variables affecting C and N stocks.

## 5. Conclusions

The fragile soils of the Tunisian oases have a low level of organic carbon and total nitrogen. Moreover, their stocks decrease rapidly under local conditions. It is very useful to follow the beneficial effect of the most suitable culture between barley (BA) alone, alfalfa (AL) alone, intercropping barley/alfalfa (BA/AL), and fallow (FA) on C and N stocks in order to draw precise conclusions.

It emerges from this study that alfalfa crops alone and especially in combination with barley could be one of the effective measures to sequester organic carbon and nitrogen and improve the BD and salinity of the soil in order to reduce the degradation of the oasis of Degache, and all the other Tunisian oases.

After 21 months, the stock under AL/AF intercropping significantly increased the stock of C and N. Therefore, a clear improvement was recorded in the BD and EC, especially in the surface layers.

This study highlighted the impact of the rhizosphere of intercropping on the enrichment of oasis soil in C and N. This finding could remedy the poverty of these soils, guarantee their fertility, improve their agricultural yields, and consequently relieve the country's economy. In fact, after almost two years, the main results obtained have shown that under fallow conditions, the organic stock decreases, and under barley cultivation, only the N stock has also decreased. Only intercropping gave the greatest increases in C and N stock than in single crops and fallows.

We can conclude that in low soils C and N stocks, intercropping could be an effective solution and could replace fallow or culture alone systems in Tunisian oases, which can improve the soil with the input of organic matter.

**Supplementary Materials:** The following are available online at https://www.mdpi.com/article/10.3390/agronomy11101903/s1, Table S1: Descriptive statistics of soil properties at the four sub-plots between 2020 and 2021 (n = 3).

**Author Contributions:** All authors contributed extensively to this work. Conceptualization, N.B.; methodology, N.B., N.K., L.D. and A.B.; software, N.B.; formal analysis, N.B., N.K., L.D. and A.B.; investigation, N.B. and L.D.; data curation, N.B., N.K., L.D. and A.B.; writing—original draft preparation, N.B. and N.K.; writing—review and editing, N.B., N.K., L.D. and A.B.; supervision, N.B. and A.B. All authors have read and agreed to the published version of the manuscript.

**Funding:** This research received no external funding.

**Institutional Review Board Statement:** Not applicable.

**Informed Consent Statement:** Not applicable.

**Data Availability Statement:** Data presented in this study are available on request from the corresponding author.

**Acknowledgments:** The authors would like to thank the Tunisian Ministry of Higher Education and Scientific Research.

**Conflicts of Interest:** The authors declare no conflict of interest.

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
