# Peer review of "Global Landscape of Organic Carbon and Total Nitrogen in the Soils of Oasis Ecosystems in Southern Tunisia"

_agronomy, doi:10.3390/agronomy11101903_

Round 1
Reviewer 1 Report
The authors tested the effect of intercropping system on oasis soils of Tunisia. They analysed C and N content in the soils cultivated with Barley alone, Alfalfa alone, intercropping Barley/Alfalfa, and a control fallow, in a saline desert soil poor in organic matter.
After two years of experiment, the best result is that of the Barley/Alfalfa inter-cropping with an increase of + 126.46% in the C stock and + 178.67% in the N stock.
This result encouraged authors to support the use of intercropping to increase productivity in the oases of Tunisia.
In my opinion, the work is interesting. It is well written. All sections are clear and satisfying.
Best regards.
Author Response
Dear Sir,
Thank you very much for your time reading and reviewing our article.
Thank you for your trust and your positive response.
Best regards
Reviewer 2 Report
- Please correct language and editing errors.
- On what basis were the doses of organic and mineral fertilization determined?
- Please add references to the all methods used.
- Please introduce every abbreviation and acronym before using it in the text (put them in parentheses after the full terms).
- All parameters should be subjected to statistical analysis to determine their statistical validity.
Author Response
Dear Sir,
Thank you very much for your time reading and reviewing our article.
You will find my answers under your questions.
Best regards.
1. Please correct language and editing errors.
The language has been revised by a specialist. See the new version of the manuscript
2. On what basis were the doses of organic and mineral fertilization determined?
See answer to your question in the new version of the manuscript, lines 190-198.
3. Please add references to the all methods used.
References are added. See new version of the manuscript.
4. Please introduce every abbreviation and acronym before using it in the text (put them in parentheses after the full terms).
We have checked all abbreviations and acronyms throughout the text. See new version of the manuscript.
5. All parameters should be subjected to statistical analysis to determine their statistical validity.
We have added at the end of the paper, after the bibliography section a supplementary table for the statistical analysis of the various parameters. See new version of the manuscript.

Reviewer 3 Report
Title: Global landscape of organic carbon and total nitrogen in the soils of oasis ecosystems in southern Tunisia
Abstract: The abstract is poorly written. I suggest it is rewritten as one unit containing the numbered subheadings (introduction, aim, hypothesis, result and conclusion) in a free flowing passage. I suggest one line of the background of study in abstract to attract the reader
Introduction:
treat = threat
I find this section well written as it gives a good background of the research in question. Also the aim of the study is evident in the beginning and concluding parts. I will only suggest the section is thoroughly revised with respect to grammar.
Materials and Methods:
I find this section well-structured and scientifically valid.
Results: The results are well presented. I will only suggest the figures are made more legible.
Discussion: This section is fairly good but will require more justifications with some more comparisons with similar studies in the past.
The manuscript is not well concluded. I suggest a revision to make capture the outcome of the study in a more concise way.
Although the study is interesting and will appeal to readers, I suggest the comments raised (from all reviewers) are duly addressed to make it more comprehensible and concise to bring the quality to a publishable level. Also grammar revision should be done.
Author Response
Comments and Suggestions for Authors
Dear Sir,
Thank you very much for your time reading and reviewing our article.
You will find my answers under your questions.
Best regards.
« Abstract: The abstract is poorly written. I suggest it is rewritten as one unit containing the numbered subheadings (introduction, aim, hypothesis, result and conclusion) in a free flowing passage. I suggest one line of the background of study in abstract to attract the reader ».
The abstract is rewritten again in response to your suggestions. See new version of the manuscript.
« Introduction:
treat = threat
I find this section well written as it gives a good background of the research in question. Also the aim of the study is evident in the beginning and concluding parts. I will only suggest the section is thoroughly revised with respect to grammar ».
The introduction and all the text has been corrected by a specialist in English. See new version of the manuscript.
« Materials and Methods:
I find this section well-structured and scientifically valid ».
Thank you!
We have improved this section by adding the bibliographic reference for each analysis performed according to the instructions of other reviewers.
See new version of the manuscript.
« Results: The results are well presented. I will only suggest the figures are made more legible ».
By enlarging the figures, their resolution improves.
« Discussion: This section is fairly good but will require more justifications with some more comparisons with similar studies in the past ».
We have improved this section. See new version of the manuscript.
« The manuscript is not well concluded. I suggest a revision to make capture the outcome of the study in a more concise way ».
The conclusion is rewritten, see new version of the manuscript.
« Although the study is interesting and will appeal to readers, I suggest the comments raised (from all reviewers) are duly addressed to make it more comprehensible and concise to bring the quality to a publishable level. Also grammar revision should be done ».
All comments and suggestions from reviewers have been taken into consideration. A reading in English by a language specialist was carried out. See the new version.
